# *FM*^2^ Path Planner for UAV Applications with Curvature Constraints: A Comparative Analysis with Other Planning Approaches

**DOI:** 10.3390/s22093174

**Published:** 2022-04-21

**Authors:** Santiago Garrido, Javier Muñoz, Blanca López, Fernando Quevedo, Concepción A. Monje, Luis Moreno

**Affiliations:** Robotics Lab, Department of Systems Engineering and Automation, Universidad Carlos III de Madrid, Av. Universidad 30, 28911 Madrid, Spain; jamunozm@ing.uc3m.es (J.M.); bllopezp@ing.uc3m.es (B.L.); fquevedo@ing.uc3m.es (F.Q.); cmonje@ing.uc3m.es (C.A.M.); moreno@ing.uc3m.es (L.M.)

**Keywords:** fast marching, Dubins, Euler–Mumford Elastica, Reeds–Shepp, curvature constraint

## Abstract

This paper studies the Fast Marching Square ($FM^{2}$) method as a competitive path planner for UAV applications. The approach fulfills trajectory curvature constraints together with a significantly reduced computation time, which makes it overperform with respect to other planning methods of the literature based on optimization. A comparative analysis is presented to demonstrate how the $FM^{2}$ approach can easily adapt its performance thanks to the introduction of two parameters, saturation α and exponent β, that allow a flexible configuration of the paths in terms of curvature restrictions, among others. The main contributions of the method are twofold: first, a feasible path is directly obtained without the need of a later optimization process to accomplish curvature restrictions; second, the computation speed is significantly increased, up to 220 times faster than other optimization-based methods such as, for instance, Dubins, Euler–Mumford Elastica and Reeds–Shepp. Simulation results are given to demonstrate the superiority of the method when used for UAV applications in comparison with the three previously mentioned methods.

## 1. Introduction

Autonomous robots and ground, marine and aerial vehicles have increased their technological level over the last 20 years, today being a mature technology in some fields and increasing their maturity level in others. Their capabilities and autonomy are continuously increasing with new applications, operating for longer periods and in more complex environments and situations. One of the factors that significantly affects the vehicle autonomy level is the path-planning ability to guarantee accurate, safe and executable trajectories. This path-planning capability is critical for most robots, and particularly challenging for UAVs.

Motion planning of robots is an extensively researched area [1]. In most domains, and for different reasons, planners take into consideration only geometric constraints. Widely used approaches for the path-planning problem are grid-based methods based on graph-search strategies to extract the optimal path, such as Dijkstra’s Algorithm [2] and A* [3]. However, these methods depend on the discretization that should balance computational requirements and accuracy, and that, on occasion, can lead to non-optimal solutions or even to failures if the grid does not have enough resolution.

A different and widely used class of algorithms are sampling-based methods, or stochastic ones, such as probabilistic roadmap (PRM) [4] and rapidly exploring random tree (RRT) [1]. These methods rely on randomly sampling configurations from the C-Space and connecting them to each other. These approaches are not limited by the resolution of the grid, as in the case of search-based methods, but are strongly limited by the sampling process. The solution will converge towards the optimal as the number of iterations goes towards infinity. A drawback of sampling-based algorithms is the lack of guarantees to find a solution and optimality.

For non-holonomic robots such as UAVs, finding a collision-free path is not sufficient: the path also needs to be executable, which means that the vehicle’s execution of the path should verify the vehicle’s motion constraints. Different variations of the previously mentioned methods have dealt with the way to take into account these constraints at planning time. Yan et al. [5] used A* with a circular search to ensure path feasibility for a UAV. Pivtoraiko et al. [6] extended A* to connect states based on Reeds–Shepp curves. Another approach to solve the path planning problem under motion constraints with a search-based method is the hybrid A* (HA*) [7]. In a similar way, different sampling-based algorithms consider the motion constraints of the robot at planning. RRT was initially developed for kinematic planning [8]. Expansive space trees (EST) [1] performs random sampling from a random state to explore the search space with kinematic constraints. Hernández et al. added Dubins curves [9] to enable RRT* to be used with kinematic constraints [10]. Stable sparse-RRT (SST) was used for AUVs by Pairet et al. [11].

However, an important question arises: if the planning method is able to generate smooth enough paths to verify the vehicle’s motion constraints, is it then necessary to introduce these constraints in the path planning method? Moreover, if we introduce them, what are the advantages and disadvantages of doing so? These questions are addressed in this work.

Other common approaches try to solve the path-planning problem include minimising the first and second derivatives of the line curvature by means of a quadratic optimisation problem [12] or solving an additional Optimal Control Problem in order to find the curvature of the centre line [13]. In contrast to these approaches, Heilmeier et al. [14] performed an approximate spline regression and generated a smooth reference line, which varies slightly from the centre line, and thus offers more scope for action in the smoothing procedure. However, these methods are not appropriate for path planning and collision risk management strategy for multi-UAVs in 3D environments [15], where maps with a grid of 1000 × 1000 × 40 or similar dimensions are used and multiple UAVs have to be planned. Our approach focuses on solving this problem avoiding the use of optimization and providing a very fast computation of the resulting paths.

Throughout the years, the Fast Marching Square ($FM^{2}$) method has demonstrated its robust performance when it comes to the calculation of the paths to be followed by UAVs during the execution of their missions. As a path planner, the method has been demonstrated to obtain trajectories which are smooth, with no sharp turns, and safe (far enough from obstacles) to be executed by ground, submarine and aerial robots. All this is achieved without the need to include the vehicle’s motion constraints in the computation of the algorithm.

In this work, a study is carried out to demonstrate how $FM^{2}$ can overperform other motion constraints based approaches such as Dubins, Euler–Mumford Elastica and Reeds–Shepp Forward methods when applied for the path planning of quadcopters and fixed-wing drones. These methods have been selected due to the fact that they are a reference in the field when it comes to optimization-based planners and many others are based on them. Aspects such as the curvature restriction of the trajectories and the computation time involved in the obtaining of the paths will be addressed.

In 1691, J. Bernoulli [16,17] formulated the classic version of the deformation problem of a flexible rod, or isotropic ideal elastic bar with a uniform elastic stiffness, which bends due to being subjected to external forces and moments applied at its ends. This is called the Euler–Mumford Elastica problem.

Dubins curves are those that connect two points with the shortest path, with the restriction that the curvature cannot exceed a given value and that the vehicle can only go forward. In 1957, Dubins [9] showed that these curves are composed of arcs of circumference with the maximum allowed curvature and line segments.

The Reeds–Shepp problem (1990) [18] is similar to that of Dubins, but the arcs can have different curvatures. Usually, the vehicle can also travel in reverse direction. However, in this paper, only the Reeds–Shepp Forward method is considered.

The study presented in this paper will compare the trajectories obtained by each of these three methods with those obtained using the $FM^{2}$ approach when varying two performance parameters, α and β, that will be defined and discussed later. In order to do this, the distance matrix will be calculated using the Fréchet distance [19] between two curves. This distance is a measure of similarity between two discrete curves, P and Q (see Figure 1). It is defined as the minimum cord-length sufficient to join a point traveling forward along P and one traveling forward along Q, although the rate of travel for either point may not necessarily be uniform. Moreover, the area between two discrete curves in an XY plane will be also used as a distance for comparison.

As a first step for comparison, Table 1 shows the computation times of the four path-planning methods of the study taken to calculate the different trajectory solutions (they will be addressed in the following sections). The time difference is of three orders of magnitude in favor of the $FM^{2}$ method, which is approximately 220 times faster than the rest. This is the reason why a deeper analysis of the results is of interest and a further study of the curvature problem will be carried out.

## 2. Description of the Planning Approaches

This section describes the four path-planning approaches studied and compared in this work, namely, $FM^{2}$, Dubins, Euler–Mumford Elastica and Reeds–Shepp. First of all, our $FM^{2}$ approach will be presented, followed by the introduction of the other three well-known planning methods.

### 2.1. FMM and $FM^{2}$ Methods

The Fast Marching method (FMM) was developed by J.A. Sethian [20] in 1996 to solve the Eikonal Equation (Equation 1), which is the equation that models the propagation of light and other electromagnetic waves in a medium with refractive index 1/F(x), where F(x) is the velocity of propagation of the medium and T(x) is the arrival time of the wave front.  
(1)|∇T|F=1

The objective of the FMM is to solve the discretized form of this equation in an orthogonal mesh. Once the equation is solved, the result is a funnel-shaped surface and the path of minimum distance can be found using the gradient method [21].

The FMM models phenomena based on the evolution of a normally self-propagating wavefront, such as the propagation of light or that of an oil slick in the sea. The following explanation is in two dimensions, but it can be extended to three or more dimensions. Let $T_{ij}$ be the time in which the wavefront crosses the point (i,j) of a two-dimensional map, satisfying |∇T|F=1, the Eikonal equation. $F=F_{ij}$ represents the velocity function of the wave at that point on the map. The main idea of the algorithm is to compute the arrival time map using only upwind values adjacent to the wavefront which, as shown in [22], leads to a simplified solution as follows:(2)$${\left(\frac{T-T_{x}}{\Delta x}\right)}^{2}+{\left(\frac{T-T_{y}}{\Delta y}\right)}^{2}=\frac{1}{{F_{ij}}^{2}},$$
where $T_{x}$ and $T_{y}$ are the minimum arrival time between the neighbors in *X* and *Y* axes, respectively, and $F_{ij}$ is the speed of propagation in the cell for which the arrival time is being computed. Equation (Equation 2) is a regular quadratic equation of the form $aT^{2}+bT+c=0$, being:(3)$$\begin{array}{c} a=\Delta_{x}^{2}+\Delta_{y}^{2},\\ b=-2(\Delta_{y}^{2}T_{x}+\Delta_{x}^{2}T_{y}),\\ c=\Delta_{y}^{2}T_{x}^{2}+\Delta_{x}^{2}T_{y}^{2}-\frac{\Delta_{x}^{2}\Delta_{y}^{2}}{F_{\mathrm{ij}}^{2}}, \end{array}$$
where, in order to simplify the notation, we assume that the grid is composed of unit square cells, that is, Δx=Δy=1.

The complete procedure for calculating the FMM solution is detailed in Algorithm 1. While running, the algorithm classifies map points into three sets: frozen, open, and unvisited. Frozen points are those for which the arrival time can no longer change. Unvisited points are those that have not been processed yet. Finally, the open points are those that can be considered as an interface between frozen and unvisited regions of the map, belonging to the propagating wave front.

In the first step of the algorithm, initialization, all cells in the map are initialized to infinity (or the maximum value possible in the computer) and are set to unvisited, except for the starting point (the destination point of the path planning), which is set with an arrival time of 0 and is considered as the first open point.
**Algorithm 1** Fast Marching Method**Require:** A grid map *X* of size m×n, source point $x_{0}$.**Ensure:** The grid map *X* with the *T* value set for all cells.*  Initialization.*1:**for all **x∈X**do**2:    T(x)←∞;3:**end for**4:$T(x_{0})\leftarrow 0;$5:$frozen\leftarrow x_{0};$6:$open\leftarrow N(x_{0});$            ▹ Neighbors of $x_{0}$.7:open←X∖(frozen∪open);*  Iteration.*8:**while**frozen≠X**do**9:    $x_{1}\leftarrow \underset{x\in open}{\arg\min}d\left(x\right);$10:    **for all** $x_{i}=N\left(x_{1}\right)\in T\cap \notin frozen$ **do**11:        $\mathrm{UPDATE}(x_{i});$12:        $open\leftarrow open\cup \{x_{i}\};$13:    **end for**14:    $open\leftarrow open\setminus \{x_{1}\};$         ▹ Updating sets.15:    $frozen\leftarrow frozen\cup \{x_{1}\};$16:**end while**

At each iteration, the open point with the smallest value of T(x) is chosen and set as frozen. Then, the arrival time of its von-Neumann neighbors (if they are not labeled as frozen) is analyzed by solving Equation (Equation 2) and checking if the calculated value is less than the real one and labeled as open (called *UPDATE* in the algorithm). This procedure continues until all points are set to frozen or the start point of the route is reached.

The minimal distance path has the problem that it can be too close to corners and has sharp turns (Figure 2). To avoid this drawback, the $FM^{2}$ method was developed [23,24,25], which consists of applying the FMM twice. The first takes as the origin of the expansion of the wave all the points corresponding to the walls and obstacles. This provides a kind of distance or first potential *W* of gray levels, or wave expansion velocities, darker near the walls and obstacles (slower) and lighter at the furthest points (faster).

On this first potential *W*, the FMM can be applied again, taking the destination point as the origin of the expansion of the wave and expanding the waves until reaching the origin point. Next, the gradient method is applied to obtain the fastest path according to the metric given by *W*. In this way, a smoother trajectory is obtained which moves away from walls and corners (Figure 3).

There are two parameters affecting the curvature of the resulting paths from the $FM^{2}$ approach:The first is the saturation parameter α, which consists of setting a maximum value, between 0 and 1, from which all the values of the matrix *W* are equal to 1.In Figure 3 and Figure 4, for α = 0.8 and α = 0.6, respectively, it can be seen that the white area away from obstacles is more or less large. A saturation value closer to zero implies less curved trajectories, but with more angle at certain points and closer to obstacles. The trajectory of Figure 3c) also presents a curvature in some points bigger than that of Figure 4c).The second parameter, β, is an exponent between 0 and 1 to which each coefficient of matrix *W* of the first potential is raised. Figure 5 and Figure 6 show the behaviour of the algorithm for varying values of β. The closer the exponent to zero, the clearer the image and the less curvature of the trajectories. The closer the exponent to one, the more similar the curvature to the Medial Axis Transform (MAT), although the smoother the trajectories and the more separated from obstacles (safer paths).It is also possible to raise each coefficient of the matrix to numbers greater than one, but the results are not of interest because they are more and more similar to the trajectories obtained with a Voronoi diagram that have sharp edges.

### 2.2. Dubins, Euler–Mumford Elastica and Reeds–Shepp methods

Mirebeau [26,27] developed numerical methods to calculate trajectories that minimize a cost function to obtain the paths corresponding to the models of the Dubins car, the Euler–Mumford Elastica and the Reeds–Shepp car using the Fast Marching approach with stencils method. For this, he discretizes the generalized Eikonal equation, also called the first-order static Hamilton–Jacobi–Bellman equation. The total cost function for a curve $x:\left[0,T\right]\to R^{\in }$, parameterized with unit velocity, is
(4)$$\int_{0}^{T}\gamma \left(x\left(s\right),\dot{x}\left(s\right)\right)C(|\ddot{x}\left(s\right)|))ds,$$
where $\gamma :R^{2}xS^{1}\to ]0,\infty [$ is a continuous cost function, which in our case is taken as the unit, and the path curvature $|\xi_{\kappa }x\left(s\right)|>0$ is penalized using a second cost function $C:R\to R_{+}$, which for the three models is
(5)$$\begin{array}{c} \begin{array}{c} C^{D}\left(\kappa \right):=\left\{\begin{array}{cc} 1, & if\;|\xi_{\kappa }|\leq 1,\\ \;+\infty , & otherwise. \end{array}\right. \end{array}\\ C^{EM}\left(\kappa \right):=1+{\left|\xi_{\kappa }\right|}^{2},\\ C^{RS}\left(\kappa \right):=\sqrt{1+|\xi_{\kappa }{|}^{2}}, \end{array}$$

The Dubins cost function penalizes only the length of the path, unless a certain threshold curvature value is exceeded, in which case the path is discarded. The minimum trajectories are obtained for γ=1. In Equation (5), $C^{D}\left(\kappa \right)$ can have only two values, 1 for the circumference arcs and +∞ for straight line segments.

The Euler–Mumford Elastica cost function has the physical meaning of the energy required to bend an elastic rod. In Equation (5), $C^{\mathrm{EM}}\left(\kappa \right)$ has the meaning of unit energy.

The Reeds–Shepp cost function is used to model slow vehicles, especially for wheelchairs. In Equation (5), $C^{\mathrm{EM}}\left(\kappa \right)$ has the meaning of square root of unit energy.

For both the Euler–Mumford Elastica and the Reeds–Shepp cost functions, the curvature changes continuously along the curve.

## 3. Discussion of Simulation Results

This section addresses the simulation results obtained when applying the four planning methods described before. First, some considerations regarding the execution of the different algorithms are given.

### 3.1. Simulation Execution Conditions

The following considerations apply for the simulations:1.The map in Figure 7, corresponding to the city area of the Centre Pompidou, is used for the simulations.2.A total of 24 different trajectories are executed by each method, covering the main central area of the map.3.For the execution of the Dubins car, the Euler–Mumford Elastica and the Reeds–Shepp approaches, the codes proposed in [26,27] by Mirebeau et al. are used, which apply the Fast Marching with stencils method (see Section 2.2 and Equations (Equation 4) and (5) for a detailed description of the curvature cost functions). The initial and end points of the 24 trajectories are specified, and parameter ξ is taken as 1.4.The $FM^{2}$ method is applied using the same initial and end points defined before, varing parameters α and β as described in the following subsection.5.For the sake of comparison of the resulting paths and their curvatures using these four methods, two measures of similarity will be used: the Fréchet distance and the area between two curves, as defined before. The results are given and discussed in the following subsections.

**Figure 7 sensors-22-03174-f007:**
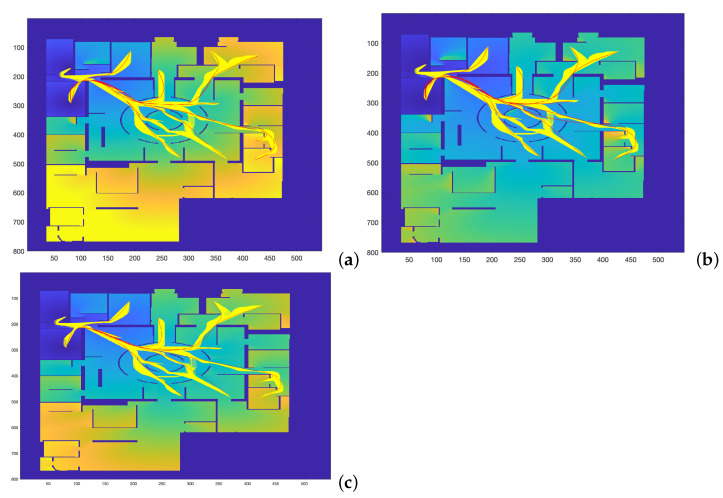
Trajectories obtained by varying α and β parameters to study which set of parameters makes the $FM^{2}$ trajectories more similar to those obtained by the (**a**) Dubins, (**b**) Euler–Mumford Elastica, and (**c**) Reeds–Shepp methods.

### 3.2. Minimization of the Fréchet Distance

In this section, the comparison of the four planning methods is addressed studying the minimization of the Fréchet distance, as described in the introduction (Figure 1).

The map used for planning is the one represented in Figure 7. This figure shows the comparison of the trajectories obtained by the $FM^{2}$ method in yellow, for α = 0.02:0.02:1.0 and β = 0.02:0.02:1.0, with those obtained by the other three methods in red. The computation times involved in the calculation of these trajectories are the ones presented and discussed in Table 1 (significantly faster computation for the $FM^{2}$ case).

As can be seen, the trajectories and their curvatures are quite similar at a first sight. The objective is to calculate the values of α and β that better approximate the Dubins, Euler–Mumford Elastica and Reeds–Shepp curves. To do that, it is necessary to have metrics for the distance between two curves. The Fréchet distance can be used for this purpose, as described previously.

The distance matrices between the $FM^{2}$ curves and the curves obtained by each of the other three planning methods are shown in Figure 8. Using these matrices, the values of α and β that make this distance minimum can be obtained. As can be seen in the three sub-figures, parameters α and β are coupled and a result similar to the Pareto fronts is obtained.

Figure 9 shows the $FM^{2}$ curves corresponding to the best α and β values in yellow and the curves from the other three methods. Figure 10 shows the comparison of the curvatures of the curves corresponding to the best α and β values for the $FM^{2}$ method (right) and those obtained by the Dubins, the Euler–Mumford Elastica, and the Reeds–Shepp methods (left, from top to bottom, respectively). The curvatures in blue correspond to the point-to-point representations, while the red ones are obtained after applying a smoothing process to the original representations. This treatment is necessary in order to obtain curvatures to be executed by real UAVs. The smoothing method used (DCT-PLS) [28] is based on a penalized least squares approach (PLS), and combines the use of the discrete cosine transform (DCT) and the generalized cross-validation, thus allowing a fast unsupervised smoothing of the data.

Table 2 shows the best values of parameters α and β and the Fréchet cost functions when $FM^{2}$ is used to approximate the Dubins, the Euler–Mumford Elastica, and the Reeds–Shepp curves.

### 3.3. Minimization of the Area between Curves

The area between two curves can also be used as a criterion to measure the distance between them, as shown in the example of Figure 11.

As the curves are not parameterized in X or Y and can take increasing or decreasing values in different sections, they are difficult to calculated by methods based on integration. Since the curves are numerically given by matrices with two rows (X and Y coordinates) and many columns (the number of points), they can be considered polygons. As the two discrete curves have the same first point and the same last point, if the order of the second curve is reversed and they are concatenated, a closed polygon with many sides is obtained whose area can be calculated by triangulation.

Applying this area concept, now the distance matrices between the $FM^{2}$ curves and the curves obtained by each of the other three planning methods are shown in Figure 12. Using these matrices, the value of α and β that make this distance minimum can be obtained. As can be seen in the three sub-figures, parameters α and β are coupled and a result similar to the Pareto fronts is obtained.

Figure 13 shows the curves corresponding to the best α and β values for the $FM^{2}$ method in yellow in comparison with the Dubins, the Euler–Mumford Elastica, and the Reeds–Shepp curves in red.

Figure 14 shows the comparison of the curvatures of the curves corresponding to the best α and β values for the $FM^{2}$ method (right) and those obtained by the Dubins, the Euler–Mumford Elastica, and the Reeds–Shepp methods (left, from top to bottom, respectively). The curvatures in blue correspond to the point-to-point representations, while the red ones are obtained after applying a smoothing process to the original ones, as discussed for the case of the Fréchet distance.

Finally, Table 3 shows the best values of parameters α and β and the area cost functions when $FM^{2}$ is used to approximate the Dubins, the Euler–Mumford Elastica, and the Reeds–Shepp curves. It can be seen that taking the area between two curves as the cost funtion for the minimization provides better results in comparison to the Fréchet distance.

### 3.4. Distance from Paths to Obstacles

In addition to the curvature of the curves, another important measure of quality of the trajectories is how close they are to obstacles, that is, the minimum distance to obstacles. Table 4 shows the distances to the obstacles of the trajectories obtained by the Dubins, the Euler–Mumford Elastica, and the Reeds–Shepp methods in comparison with the distances obtained using their best $FM^{2}$ approximations. The area cost function is used for the comparison. It can be seen that the $FM^{2}$ method provides safer curves since their distances to obstacles is significantly bigger than the others, specially for the Dubins and Reeds–Shepp cases.

### 3.5. Simulation Results Using a Quadcopter Model and a Fixed-Wing UAV

So far the study has focused on the quality of the trajectories based on their curvatures and their distances to the obstacles. However, it is also important to consider the vehicles dynamics and check if they are able to execute the paths, specially for the case of fixed-wing UAVs. In order to do this analysis, a vehicle dynamic model has been considered using the Matlab Simulink: file shared_uav_aeroblks/UAVFidelityExample [29] to simulate the tracking of the trajectories, as shown in Figure 15.

In the Simulink model, the UAV Waypoint Follower block stands out, which considers the dynamics of the UAV and allows the selection of a fixed-wing drone and a quadcopter. This block is based on a Pure Pursuit algorithm that follows a point on the desired trajectory. There are two parameters to select: the lookahead distance and the heading controller gain. The results are shown in Figure 16 for a lookahead distance of 40 m and a heading controller gain of 2, considering the $FM^{2}$ path that better approximate to the Dubins path (as calculated in the previous section). The figure shows that the difference between the reference path ($FM^{2}$ trajectory) and the executed path (UAV trajectory) is negligible enough so as to consider that the path is feasible to be used in practice.

## 4. Conclusions

From the results presented in this work, some important conclusions can be reached. First of all, the $FM^{2}$ method has demonstrated to provide paths whose smoothness, curvature and distance to the obstacles can be easily adjusted thanks to the use of parameters α and β. Through the minimization of functions such as the Fréchet distance between two curves or the area between them, the tuning of these two parameters can be performed so that the resulting paths are competitive in comparison to those provided by other planning approaches such as the Dubins, the Euler–Mumford Elastica, and the Reeds–Shepp methods.

The results show that selecting the area between two curves as the cost function for the minimization provides better results than the Fréchet distance when it comes to approximating the $FM^{2}$ trajectories to those of the other three methods. In any case, other cost functions could be studied, and the alternative of selecting the curvature instead of the curve itself for the minimization could be of interest. These issues will be addressed in future works.

On the other hand, simulation tests have shown that the resulting paths are feasible and can be executed by both quadcopters and fixed-wing UAVs (more demanding dynamics), which allows the $FM^{2}$ method to be used for real drones applications.

Moreover, the comparison study has shown that the $FM^{2}$ method significantly overperforms the others in respect to computation time, computing the trajectories around 220 times faster. The nature of the algorithm and the fact that the vehicle’s kinematic constraints do not need to be introduced in the computation process allow this fact. It also opens the possibility to use this method in real time for dynamic path-planning applications where the environment is changing during the planning activity (moving obstacles or other UAVs flying in the map).

## Figures and Tables

**Figure 1 sensors-22-03174-f001:**
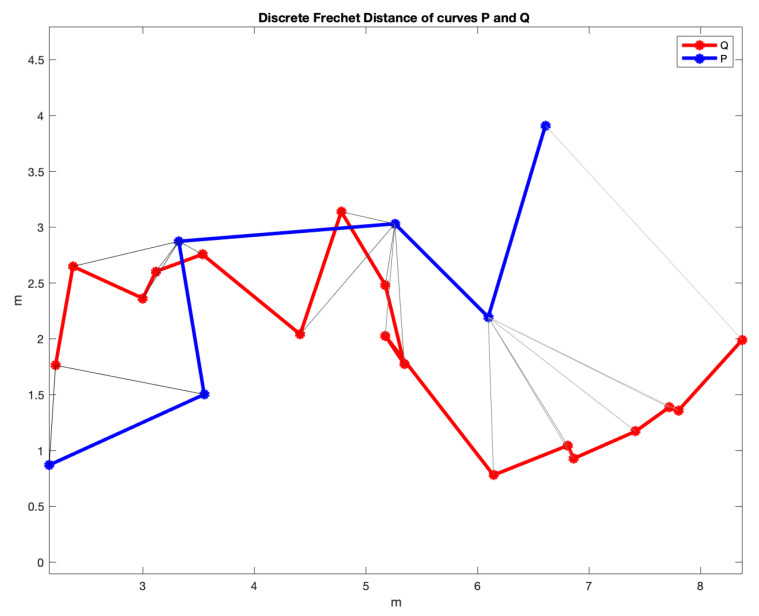
The Fréchet distance between two curves is the maximum length of all the gray lines.

**Figure 2 sensors-22-03174-f002:**
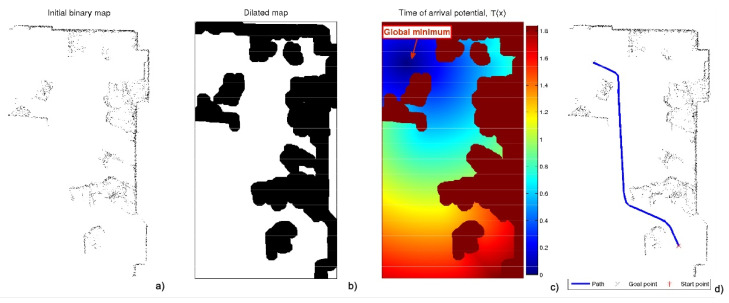
Process of obtaining the FMM trajectory. (**a**) Raw data obtained by the laser sensor; (**b**) enlargement of the previous subfigure; (**c**) expansion of the wave starting from the destination point; (**d**) path obtained by the gradient method over the potential corresponding to the previous subfigure.

**Figure 3 sensors-22-03174-f003:**
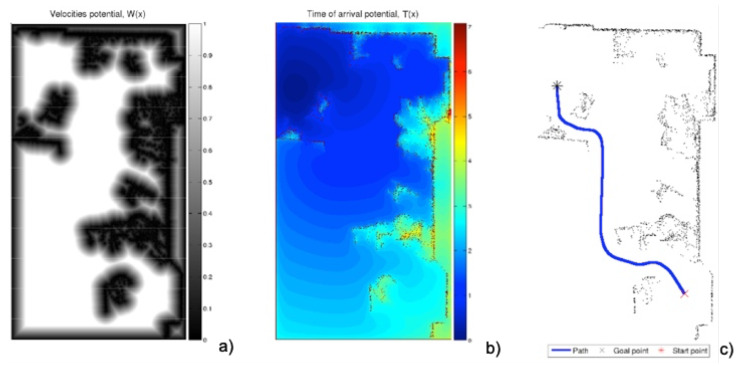
Process of obtaining the $FM^{2}$ trajectory with the same raw data obtained by the laser sensor of Figure 1. (**a**) First potential W(x) obtained with the expansion of the wave from the walls and obstacles, saturated with α=0.8; (**b**) expansion of the wave starting from the destination point; (**c**) path obtained by the gradient method over the potential corresponding to the previous subfigure.

**Figure 4 sensors-22-03174-f004:**
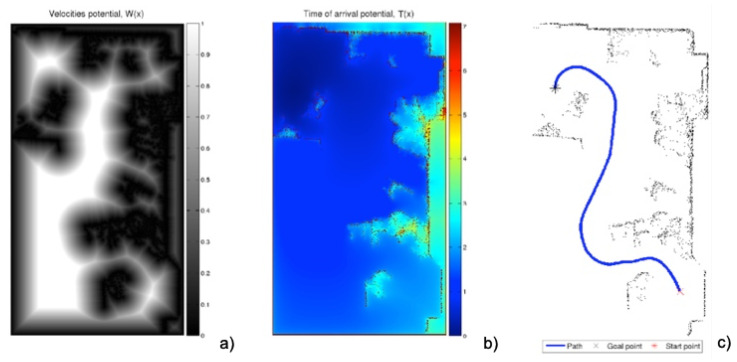
Process of obtaining the $FM^{2}$ trajectory with the same raw data obtained by the laser sensor of Figure 1. (**a**) First potential W(x) obtained with the expansion of the wave from the walls and obstacles, saturated with α=0.6; (**b**) expansion of the wave starting from the destination point; (**c**) path obtained by the gradient method over the potential corresponding to the previous subfigure.

**Figure 5 sensors-22-03174-f005:**
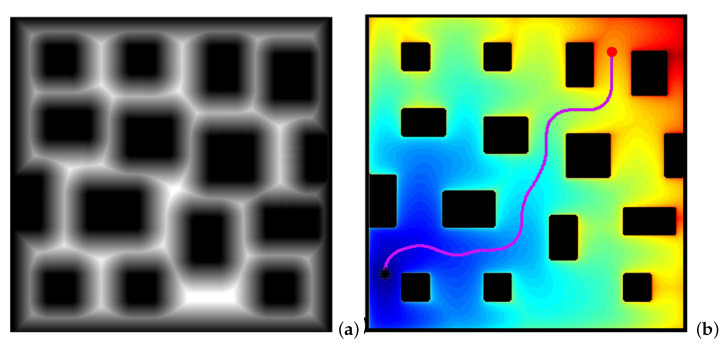
Obtaining $FM^{2}$ path in a city environment. (**a**) First potential W(x); (**b**) second potential and the path obtained with the gradient method.

**Figure 6 sensors-22-03174-f006:**
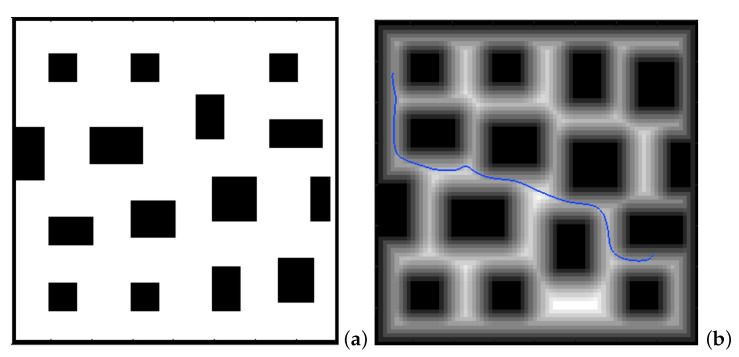
Two $FM^{2}$ paths in a city environment for different values of the exponent β of the first potential W(x). (**a**) City environment; (**b**) first potential W(x) and the path for β=0.9; (**c**) first potential W(x) and the path for β=0.4.

**Figure 8 sensors-22-03174-f008:**
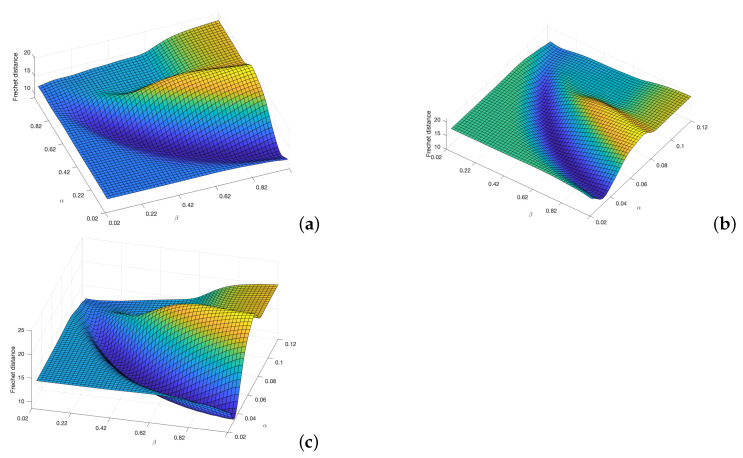
Cost functions (Fréchet distance) obtained by varying α and β parameters to study which set of parameters makes the $FM^{2}$ trajectories more similar to those obtain by the (**a**) Dubins, (**b**) Euler–Mumford Elastica, and (**c**) Reeds–Shepp methods.

**Figure 9 sensors-22-03174-f009:**
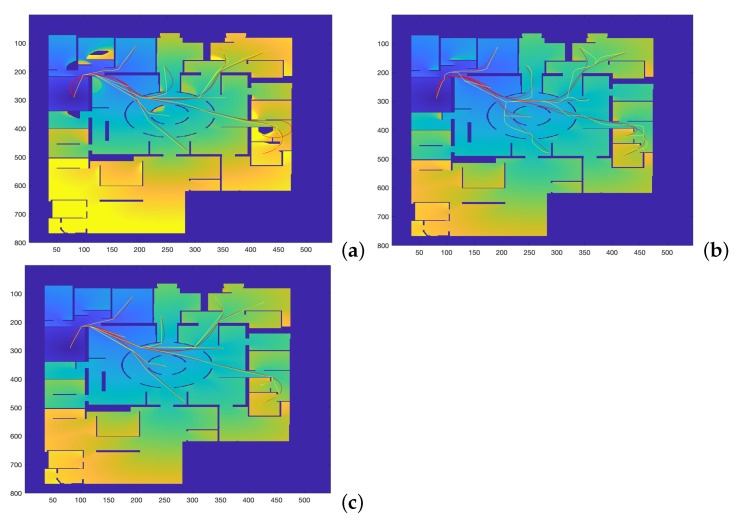
Curves corresponding to the best α and β values for the $FM^{2}$ method in yellow in comparison with the (**a**) Dubins, (**b**) Euler–Mumford Elastica, and (**c**) Reeds–Shepp curves in red. The Fréchet distance between two curves is used as cost function.

**Figure 10 sensors-22-03174-f010:**
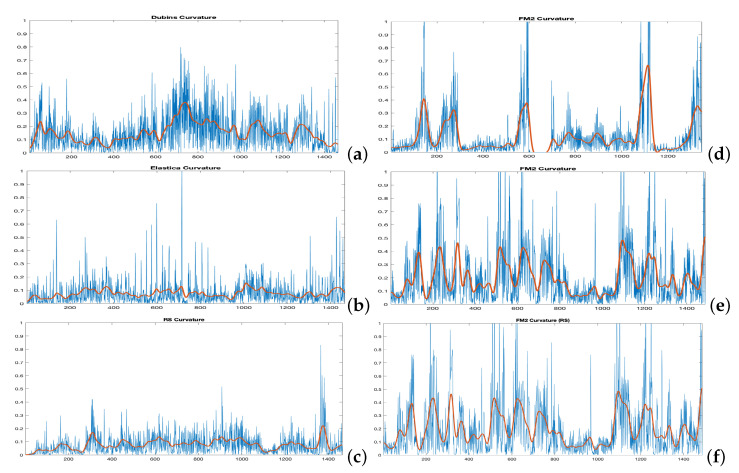
Curvatures corresponding to the (**a**) Dubins, the (**b**) Euler–Mumford Elastica, and the (**c**) Reeds–Shepp curves (left, from top to bottom, respectively) and their corresponding best approximations (**d**–**f**) using $FM^{2}$ method (right). The Fréchet cost function is used for comparison. The Fréchet distance between two curves is used as cost function. The curvatures in red are obtained after applying a smoothing process to the raw curvature data.

**Figure 11 sensors-22-03174-f011:**
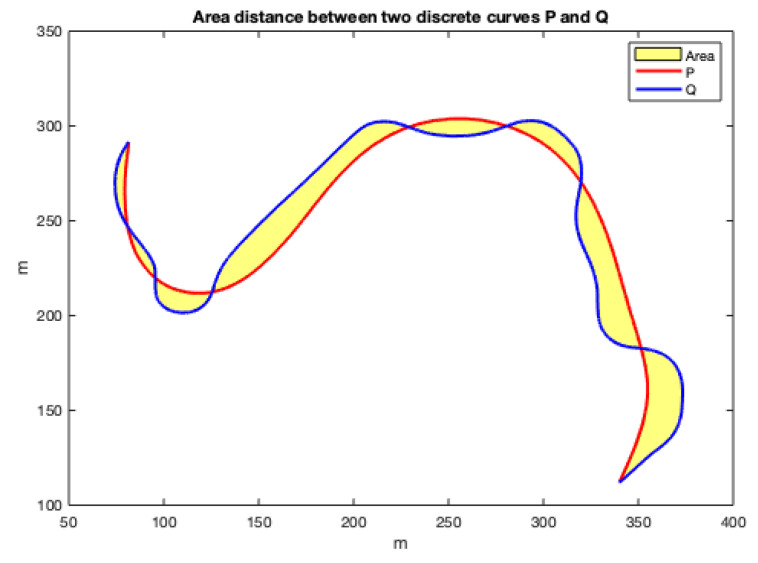
Area between two curves as a metric to measure the distance between the curves.

**Figure 12 sensors-22-03174-f012:**
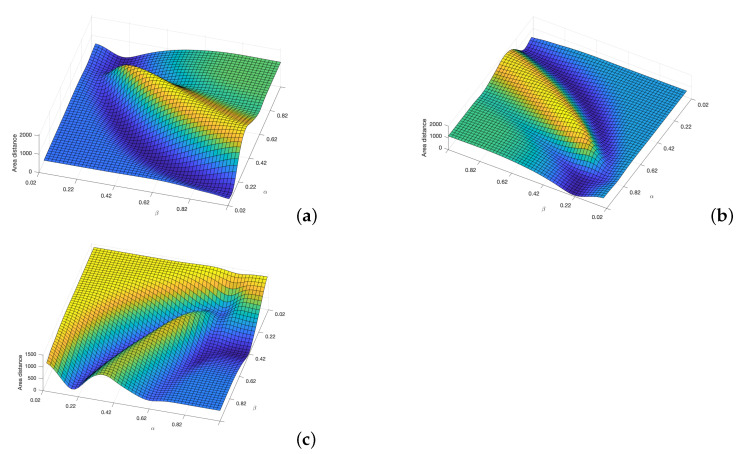
Cost functions (area between two curves) obtained by varying α and β parameters to study which set of parameters makes the $FM^{2}$ trajectories more similar to those obtain by the (**a**) Dubins, (**b**) Euler–Mumford Elastica, and (**c**) Reeds–Shepp methods.

**Figure 13 sensors-22-03174-f013:**
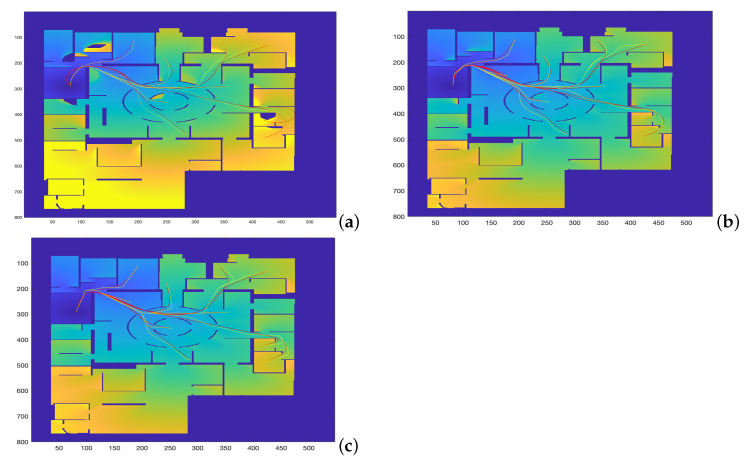
Curves corresponding to the best α and β values for the $FM^{2}$ method in yellow in comparison with the (**a**) Dubins, (**b**) Euler–Mumford Elastica, and (**c**) Reeds–Shepp curves in red. The area between the two curves is used as cost function.

**Figure 14 sensors-22-03174-f014:**
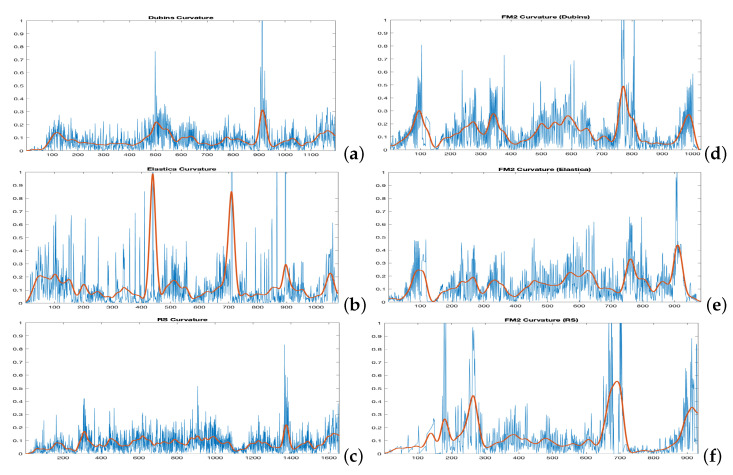
Curvatures corresponding to the (**a**) Dubins, (**b**) Euler–Mumford Elastica, and (**c**) Reeds–Shepp curves (left, from top to bottom, respectively) and their corresponding best approximations (**d**–**f**) using $FM^{2}$ method (right). The area between the curves is used as cost function for comparison. The curvatures in blue correspond to the point to point representations. The curvatures in red are obtained after applying a smoothing process.

**Figure 15 sensors-22-03174-f015:**
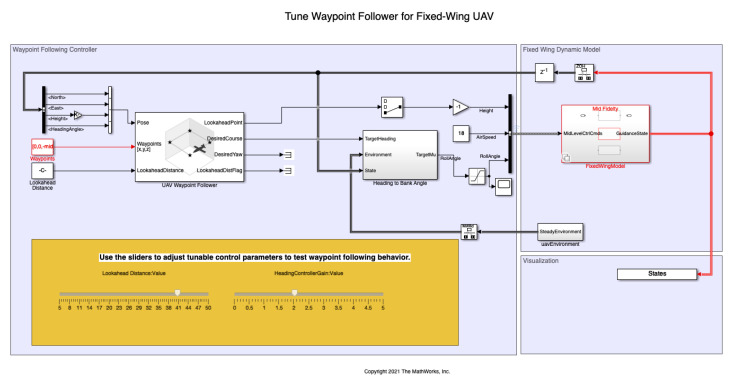
Matlab Simulink model to simulate the influence of the vehicle dynamics when executing the paths.

**Figure 16 sensors-22-03174-f016:**
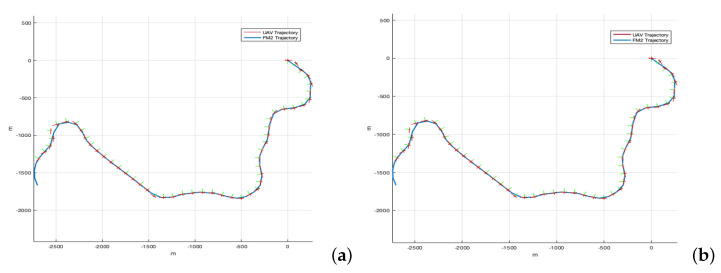
Original paths (approximation of $FM^{2}$ to Dubins method) versus executed paths considering (**a**) a fixed-wing UAV and (**b**) a quadcopter.

**Table 1 sensors-22-03174-t001:** Computation times (in seconds) of the four path planning methods, Dubins, Euler–Mumford Elastica, Reeds–Shepp and $FM^{2}$, taken to calculate the 20 trajectories addressed in the paper.

	Dubins	Euler–Mumford Elastica	Reeds–Shepp	$FM^{2}$
Computation time (secs)	18.936	19.199	19.577	0.087

**Table 2 sensors-22-03174-t002:** Best values of parameters α and β and the Fréchet cost functions when $FM^{2}$ is used to approximate the Dubins, the Euler–Mumford Elastica, and the Reeds–Shepp curves.

	Dubins	Euler–Mumford Elastica	Reeds–Shepp
α	0.1	0.3	0.038
β	0.98	0.96	0.92
Fréchet cost function	8.1	11.1	9.2

**Table 3 sensors-22-03174-t003:** Best values of parameters α and β and the area cost functions when $FM^{2}$ is used to approximate the Dubins, the Euler–Mumford Elastica, and the Reeds–Shepp curves.

	Dubins	Euler–Mumford Elastica	Reeds–Shepp
α	0.90	0.84	0.40
β	0.18	0.20	0.50
Area cost function	4.50	2.19	0.3352

**Table 4 sensors-22-03174-t004:** Distances to the obstacles of the trajectories obtained by the Dubins, the Euler–Mumford Elastica, and the Reeds–Shepp methods in comparison with the distances obtained using their best $FM^{2}$ approximations. The area cost function is used for comparison.

	Dubins	Euler–Mumford Elastica	Reeds–Shepp
α	0.90	0.84	0.40
β	0.18	0.20	0.50
Original distance	0.0093	0.0169	0.0071
Distance of the $FM^{2}$ approximation	0.0276	0.0191	0.0214

## Data Availability

Not applicable.

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
