# Peer review of "FM2 Path Planner for UAV Applications with Curvature Constraints: A Comparative Analysis with Other Planning Approaches"

_sensors, 2022, doi:10.3390/s22093174_

Round 1
Reviewer 1 Report
Manuscript has potential to be accepted in present form. Minor English editing is required.
Author Response
Reviewer 1:
Manuscript has potential to be accepted in present form. Minor English editing is required.
The authors really appreciate that this reviewer sees the potential of our work. The paper has been improved, emphasizing its main contributions and improving the English.
Reviewer 2 Report
In this paper, the authors propose the Fast Marching Square (FM2) method as a competitive path planner for UAV applications. In addition, the approach fulfills trajectory curvature constraints together with a significantly reduced computation time, which makes it overperform with respect to other planning methods of the literature.
To improve this work, the authors need to address the following issues
1. The writing skills for this papers are quite poor. The paper needs a thorough proofreading and most of the sentences need to be rephrased.
2.Some figures need to be improved. For instance, on figure number 15, the content is not clear or visible. Therefore, both figure number 15 and 16 need to be enhanced.
- The motivation for this paper and the contribution are not well explained. The literature provides other planning methods. How the authors need to answer the following concerns?
a) What is exactly and clearly the problem with other references that used those other planning methods. Why did the authors choose FM2?
b) The discussion of relevant related is not deep. We do not see the issues with relevant work.
The literature need to provide clearly the contribution of other author if possible it is good to have a comparison table on keywords.
Author Response
Reviewer 2:
In this paper, the authors propose the Fast Marching Square (FM2) method as a competitive path planner for UAV applications. In addition, the approach fulfills trajectory curvature constraints together with a significantly reduced computation time, which makes it overperform with respect to other planning methods of the literature.
To improve this work, the authors need to address the following issues
- The writing skills for this papers are quite poor. The paper needs a thorough proofreading and most of the sentences need to be rephrased.
The authors appreciate this reviewer’s suggestion. The paper has been proofread and the English has been reviewed.
- Some figures need to be improved. For instance, on figure number 15, the content is not clear or visible. Therefore, both figure number 15 and 16 need to be enhanced.
Figures have been enhanced to better show the results.
- The motivation for this paper and the contribution are not well explained. The literature provides other planning methods. How the authors need to answer the following concerns?
- a) What is exactly and clearly the problem with other references that used those other planning methods. Why did the authors choose FM2?
- b) The discussion of relevant related is not deep. We do not see the issues with relevant work.
The authors really appreciate this comment from the reviewer. The motivation and the contributions of this work haven been emphasized in the abstract and in the introduction section of the paper, in red.
In general terms, our approach fulfills trajectory curvature constraints together with a significantly reduced computation time, which makes it overperform with respect to other planning methods of the literature based on optimization. A comparative analysis is presented to demonstrate how the FM2 approach can easily adapt its performance thanks to the introduction of two parameters, saturation $\alpha$ and exponent $\beta$, that allow a flexible configuration of the paths in terms of curvature restrictions, among others. The main contributions of the method are twofold: first, a feasible path is directly obtained without the need of a later optimization process to accomplish curvature restrictions; second, the computation speed is significantly increased, up to 220 times faster than other optimization-based methods such as, for instance, Dubins, Euler-Mumford Elastica and Reeds-Shepp. Simulation results are given to demonstrate the superiority of the method when used for UAV applications in comparison with the three previously mentioned methods.
The literature need to provide clearly the contribution of other author if possible it is good to have a comparison table on keywords.
The state of the art has been improved in the introduction section, including other relevant approaches. The contributions of other authors are provided, and we have emphasized how our approach differs from them.
Reviewer 3 Report
Various parts of the manuscript needs to be improved:
- Abstract of the manuscript should be extended. More specifically, in the beginning of the abstract general idea of the research should be explained more in detail.
- The general idea of the manuscript (main aim of the work) is unclear. First of all, what actual research problem authors are trying to solve? Secondly, what is the point of taking into consideration such classical approaches as Dubins, Euler-Mumford Elastica and Reeds-Shepp, when in various research papers a numbers of intelligent and more sophisticated approaches exist? All in all, main idea of the manuscript lacks of more significant support. This is a major drawback of the manuscript.
- What is actual author contribution to the approach, described in the sub-section 2.1? Authors contribution should be highlighted more clearly.
- Structure of the manuscript is unclear and should be reconsidered. Current structure of the manuscript is misleading. For example, section 2 described the proposed approach, section 3 provides the results. Results of what? Comparison (simulations, experimental tests, etc.) procedure is not described at all. I would recommend to use more classical structure of the manuscript: 1. Introduction / literature review; 2. Development of the approach; 3. Description of the simulations / experimental tests; 4. Results and Discussion; 5. Conclusions.
- Why figures are provided in the conclusions?
- Analysis of the results and comparison should be performed in significantly more detailed way.
Author Response
Reviewer 3:
Various parts of the manuscript needs to be improved:
Abstract of the manuscript should be extended. More specifically, in the beginning of the abstract general idea of the research should be explained more in detail.
The authors thank the reviewer for this comment. The abstract has been rewritten and extended to better address the objective of the paper. Changes are remarked in red color.
The general idea of the manuscript (main aim of the work) is unclear. First of all, what actual research problem authors are trying to solve? Secondly, what is the point of taking into consideration such classical approaches as Dubins, Euler-Mumford Elastica and Reeds-Shepp, when in various research papers a numbers of intelligent and more sophisticated approaches exist? All in all, main idea of the manuscript lacks of more significant support. This is a major drawback of the manuscript.
The abstract and the introduction section of the paper have been enhanced to better present the motivation and contributions of our work.
As stated at the beginning of the introduction section, this work deals with the path planning problem when applied to UAVs and proposes a method that deals with two main issues to address when planning the paths: 1) the motion restrictions of the vehicles that execute the path, being the curvature of the path an important concern; 2) the computation time required for the obtaining of the paths. There is a wide number of approaches solving the planning problem and a significant number of them, as discussed in the introduction, are motion constraints-based, imposing curvature restrictions through optimization processes. These methods provide feasible paths, but the computation time involved in the process is significant and can compromise the planning when there is a high number of UAVs to manage and the map dimension is considerable.
In this work a study is carried out to demonstrate how FM2 can overperform other motion constraints-based approaches such as Dubins, Euler-Mumford Elastica and Reeds-Shepp Forward methods when applied for the path planning of quadcopters and fixed wing drones. These methods have been selected due to the fact that they are a reference in the field when it comes to optimization-based planners and many others are based on them. Aspects such as the curvature restriction of the trajectories and the computation time involved in the obtaining of the paths are addressed.
What is actual author contribution to the approach, described in the sub-section 2.1? Authors contribution should be highlighted more clearly.
Following the reviewer’s previous recommendation, the actual contribution of our approach has been discussed in the introduction section, as mentioned before. Section 2.1 introduces the method itself and describes how it allows us to achieve feasible paths thank to the introduction of the saturation and exponent parameters. This method was originally proposed by the authors, as shown in references [23], [24] and [5], and has been enhanced here with the introduction of these two parameters.
Structure of the manuscript is unclear and should be reconsidered. Current structure of the manuscript is misleading. For example, section 2 described the proposed approach, section 3 provides the results. Results of what? Comparison (simulations, experimental tests, etc.) procedure is not described at all. I would recommend to use more classical structure of the manuscript: 1. Introduction / literature review; 2. Development of the approach; 3. Description of the simulations / experimental tests; 4. Results and Discussion; 5. Conclusions.
The sections of the paper have been renamed using a more classical structure of the manuscript for a better understanding of each section content. Besides, an introductory text has been included at the beginning of sections 2 and 3. Regarding section 3, it is separated in three sub-sections to better address the comparison of the simulation results attending to the three different distance measures applied.
Why figures are provided in the conclusions?
It was a compilation issue and now all figures are placed out of the conclusions section.
Analysis of the results and comparison should be performed in significantly more detailed way.
As stated before, section 3 corresponding to the discussion of simulation results has been better introduced and is separated in three sub-sections to better address the comparison of the simulation results attending to the different distance measures applied. A wide number of figures and tables are provided in each sub-section for the detailed comparison of the methods.
The authors really thank the reviewers for their useful comments. They all have been considered and included in the new version of the paper and remarked in red for an easier clarification of the changes carried out.
Round 2
Reviewer 2 Report
The comments were addressed accordingly.
Author Response
The comments were addressed accordingly.
The authors really appreciate that this reviewer sees the potential of our work and supports its publication.
Reviewer 3 Report
The manuscript was significantly improved and now the mind-flow of the manuscript is more clear. However, in my opinion, couple of minor things still needs to be reconsidered:
- The structure of the manuscript is still a bit misleading. It looks like that from a structure point of view, one section (or sub-section) is still missing. As it is now, in the current version of the manuscript, section 2 describes the planning approaches. Respectively, section 3 describes the results of the simulations. Thus, the question is – where the procedure of the simulations are described, i.e. conditions, boundaries, etc.? All of the information, regarding how the simulations were performed, should be provided in more clear manner.
- How exactly the analysis part was improved?
Author Response
The structure of the manuscript is still a bit misleading. It looks like that from a structure point of view, one section (or sub-section) is still missing. As it is now, in the current version of the manuscript, section 2 describes the planning approaches. Respectively, section 3 describes the results of the simulations. Thus, the question is – where the procedure of the simulations are described, i.e. conditions, boundaries, etc.? All of the information, regarding how the simulations were performed, should be provided in more clear manner.
The authors really thank the reviewer for this valuable comment. A new subsection 3.1 has been included in the paper to better describe the simulation execution conditions. Please see the new text remarked in red color in the paper.
How exactly the analysis part was improved?
Subsections 3.2, 3.3 and 3.4 present and discuss in detail the main results obtained from the execution of each planning method. The analysis of results addresses how these four approaches compare using three main distances: the Fréchet distance (section 3.2), the area between two curves distance (section 3.3) and the distance to obstacles (section 3.4). Curves and curvatures are comparted for each case, which are the two main path parameters to study, together with the execution time of the paths. A total of 10 figures and 3 tables show the results.
The authors really thank the reviewers for their useful comments. They all have been considered and included in the new version of the paper and remarked in red for an easier clarification of the changes carried out.
The manuscript has been resubmitted to Sensors. We look forward to your positive response.
Sincerely,
Santiago Garrido.